



# Radiative effects of long-range-transported Saharan air layers as determined from airborne lidar measurements

Manuel Gutleben[1], Silke Groß[1], Martin Wirth[1], and Bernhard Mayer[2]

[1]Deutsches Zentrum für Luft- und Raumfahrt, Institut für Physik der Atmosphäre, Oberpfaffenhofen, Germany
[2]Ludwig-Maximilians-University (LMU), Meteorological Institute, Munich, Germany

**Correspondence:** Manuel Gutleben (manuel.gutleben@dlr.de)

**Abstract.** The radiative effect of long-range-transported Saharan air layers is investigated on the basis of simultaneous airborne high spectral resolution and differential absorption lidar measurements in the vicinity of Barbados. Within the observed Saharan air layers increased water vapor concentrations compared to the dry trade wind atmosphere are found. The measured profiles of aerosol optical properties and water vapor mixing ratios are used to characterize the atmospheric composition in radiative

transfer calculations, to calculate radiative effects of moist Saharan air layers and to determine radiative heating rate profiles. An analysis based on three case studies reveals that the observed enhanced amounts of water vapor within Saharan air layers have a much stronger impact on heating rate calculations than mineral dust aerosol. Maximum mineral dust short-wave heating and long-wave cooling rates are found in altitudes of highest dust concentration (short-wave: +0.5 K d$^{-1}$, long-wave: -0.2 K d$^{-1}$, net: +0.3 K d$^{-1}$). However, when considering both aerosol concentrations and measured water vapor mixing ratios in radiative

transfer calculations the maximum heating/cooling rates shift to the top of the dust layer (short-wave: +2.2 K d$^{-1}$, long-wave: -6.0 to -7.0 K d$^{-1}$, net: -5.0 to -4.0 K d$^{-1}$). Additionally, the net-heating rates decrease with height - indicating a destabilizing effect in the dust layers. Long-wave counter radiation of Saharan air layers is found to reduce cooling at the top of the subjacent marine boundary layers and might lead to less convective mixing in these layers. The overall short-wave radiative effect of mineral dust particles in Saharan air layers indicates a maximum magnitude of -40 W m$^{-2}$ at surface level and a maximum of

-25 W m$^{-2}$ at the top of the atmosphere.

## 1   Introduction

Each year during the northern hemispheric summer months from June to August large amounts of Saharan mineral dust particles are transported from the African continent towards the Caribbean islands (Moulin et al., 1997). Dust particles are injected into the atmosphere over the Saharan desert, e.g. by nocturnal low-level jets or convective activity (Schepanski et al.,

2009) and form well-mixed convective aerosol layers from ground level to up to 4-6 km height (Esselborn et al., 2008; Ben-Ami et al., 2009; Ansmann et al., 2011). Easterlies subsequently transport the layers westwards over hot desert surfaces towards the Atlantic Ocean. At the Atlantic coast the layers are undercut and lifted by cooler North Atlantic air masses and form elevated layers, the so-called Saharan Air Layers (SALs; Carlson and Prospero, 1972; Prospero and Carlson, 1972).



Embedded in the trade wind flow SALs remain relatively undisturbed and can be transported over thousands of kilometers towards the Caribbean or Americas (Carlson and Prospero, 1972; Karyampudi and Carlson, 1988; Karyampudi et al., 1999). Thus, Saharan mineral dust layers cannot be understood as a local phenomena close to their source regions. They have an impact on the radiation budget as well as on the formation and physical properties of clouds over large areas far away from

their origin.

Optical and microphysical properties of long-range-transported Saharan mineral dust (Groß et al., 2015; Haarig et al., 2018; Toledano et al., 2019) were investigated during the SALTRACE field campaign (Saharan Aerosol Long-range Transport and Aerosol-Cloud-Interaction Experiment; Weinzierl et al., 2017). Studies that aimed to quantify the radiative effects of SALs concentrated on regions near Africa at the beginning of transatlantic transport (Li et al., 2004; Zhu et al., 2007; Kanitz et al.,

2013). However, detailed studies regarding radiative effects of long-range-transported SALs are missing.

Gutleben et al. (2019a) investigated a SAL upstream the Caribbean island of Barbados using combined High Spectral Resolution Lidar (HSRL) and water vapor Differential Absorption Lidar (DIAL) technique. We found enhanced water vapor mixing ratios within the SAL compared to the surrounding dry free troposphere. From radio-soundings close to the source region we inferred an enhancement of water vapor within the SAL already at the beginning of its long-range transport. Radiosonde

measurements and Raman lidar measurements conducted by Jung et al. (2013) and Kanitz et al. (2014) support these findings. They both show enhanced water vapor concentrations as high as $5\,\mathrm{g\,kg^{-1}}$ within SALs.

Water vapor represents the Earth's strongest greenhouse gas and model calculations indicate that especially after long-range transport SAL-heating rates are highly sensitive to the used water vapor profile (Wong et al., 2009). However, previous studies which focused on the radiative effect of SALs close to their source regions lacked simultaneous measurements of water vapor

and aerosol optical properties and had to assume the water vapor vertical distribution from standard atmospheres or model simulations. Gutleben et al. (2019) however, showed that radiative heating caused by SAL-water vapor has a much greater magnitude than radiative heating by SAL-mineral dust particles and might alter the radiative heating rate profile from the bottom to the top of the SAL. Moreover, SAL-water vapor is not only influencing the SAL-thermodynamic structure itself but potentially has an impact on surrounding atmospheric layers.

Differences in shallow marine cloudiness between dust-free and dust-laden North Atlantic trade wind regions were found by Gutleben et al. (2019b), with the dust-laden regions containing less, shallower and smaller clouds. We conjectured that differences in radiative transfer could cause the observed changes. A suppressing characteristic of SAL on convection was also found by Wong and Dessler (2005), who showed that the convection barrier increases with increasing aerosol optical depth of the SAL. However, up to now it is still not understood how changes in radiative transfer could modify shallow marine

cloudiness in dust-laden trade wind regions.

The aim of this study is to investigate the radiative effect of long-range-transported SALs and to study their impact on the subjacent boundary layer. For this purpose, simultaneous measurements of water vapor concentrations and aerosol optical properties by an airborne lidar system during the Next-generation Aircraft Remote-Sensing for Validation Studies-II (NARVAL-II; Stevens et al., 2019) in August 2016 are used to characterize the vertical structure of both aerosols and water vapor mixing

ratio from flight level down to the surface. The retrieved lidar profiles are utilized for the calculation of radiative heating rate



profiles of both aerosols and water vapor using the radiative transfer model libRadtran (Library Radiative Transfer; Mayer and Kylling, 2005; Emde et al., 2016). Moreover, the radiative effect of long-range-transported SALs at surface level as well as at the top of the atmosphere (TOA) is investigated.

This paper is structured as follows. In Section 2 an overview of the NARVAL-II campaign, the applied instruments and the

used radiative transfer model setup is given. Section 3 shows the results of radiative transfer calculations for three representative case studies during NARVAL-II with different vertical dust layer extents and optical thicknesses. Results are discussed in Section 4 and focus on the radiative impact of long-range-transported SALs on the subjacent marine boundary layer (MBL) and atmospheric stability. Section 5 concludes this paper and gives a short summary.

## 2  Methods

### 2.1  The NARVAL-II field campaign

The Next-generation Aircraft Remote-Sensing for Validation Studies-II (NARVAL-II) took place in August 2016 and aimed at studying the subtropical North Atlantic atmospheric circulation using a combination of airborne remote sensing instruments (Stevens et al., 2019). Grantley Adams International Airport on Barbados (TBPB, $13°\ 04'\ 29"$ N, $59°\ 29'\ 33"$ W) served as air base for 10 measurement flights over the subtropical North Atlantic Ocean with the German High Altitude and LOng range

research aircraft HALO (Krautstrunk and Giez, 2012). During the campaign the aircraft was equipped with the lidar system WALES (Wirth et al., 2009) and a set of active and passive remote sensing instruments, i.e. the HALO Microwave Package HAMP (Mech et al., 2014; Ewald et al., 2019), the hyper spectral cloud and sky imager specMACS (Ewald et al., 2016) and the Spectral Modular Airborne Radiation measurement System SMART (Wendisch et al., 2001). In addition, a total number of 218 dropsondes were launched during the flights for measurements of the atmospheres thermodynamic state.

### 2.2  The WALES instrument

WALES is an airborne lidar system which was developed at the Institute for Atmospheric Physics of the German Aerospace Center (Wirth et al., 2009). The system is designed as a Differential Absorption Lidar (DIAL) for measurements of atmospheric water vapor distributions and operates at four different wavelengths around the water vapor absorption bands at $935\,\mathrm{nm}$. In addition to the DIAL capabilities, WALES is equipped with a polarization sensitive High Spectral Resolution Lidar (HSRL)

channel at $532\,\mathrm{nm}$ for cloud and aerosol characterization (Esselborn et al., 2008). As a result, WALES is able to perform simultaneous measurements of particle extinction coefficients ($\alpha_{\mathrm{p}(532)}$), backscatter ratios ($R_{532} = 1 + \beta_{\mathrm{p}(532)}/\beta_{\mathrm{m}(532)}$ - with $\beta_{\mathrm{p}(532)}$ and $\beta_{\mathrm{m}(532)}$ being the the particle and molecular backscatter coefficients) and particle linear depolarization ratios ($\delta_{\mathrm{p}(532)}$) at $532\,\mathrm{nm}$ as well as water vapor mass mixing ratios ($r_{\mathrm{m}}$) from flight altitude to surface level. DIAL and HSRL measurements are temporally averaged for noise reduction. As a result horizontal resolutions are of approximately $0.2\,\mathrm{km}$

for HSRL measurements and $3.0\,\mathrm{km}$ for DIAL measurements at typical aircraft speeds around $200\,\mathrm{m\,s^{-1}}$. WALES measures in near nadir direction ($2° - 3°$ off-nadir angle) with vertical resolutions of $15\,\mathrm{m}$. Pulse repetition rates of $10\,\mathrm{ms}$ between





**Table 1.** Vertical resolution of the radiative transfer model.

| Altitude [km] | Resolution [km] |
| --- | --- |
| 0.0 to 10.0 | 0.1 |
| 10.0 to 30.0 | 1.0 |
| 30.0 to 60.0 | 10.0 |
| 60.0 to 120.0 | 20.0 |

online and offline pulses allow high quality water vapor measurements with relative uncertainties of less than 5 % (Kiemle et al., 2008). Relative uncertainties of backscatter, particle linear depolarization and extinction measurements sum up to 5 %, 10-16 %, and 10-20 %, respectively (Esselborn et al., 2008).

### 2.3 The radiative transfer model libRadtran

5 Calculations of both downward and upward irradiances as well as atmospheric heating rates are performed utilizing the radiative transfer equation solver DISORT (Stamnes et al., 1988) with an improved intensity correction (Buras et al., 2011). The solver is embedded in the Library Radiative Transfer model (libRadtran; Mayer and Kylling, 2005; Emde et al., 2016) and is applied with 16 streams in the short-wave (0.12-4.0 $\mu$m) and long-wave (2.5-100.0 $\mu$m) spectra. At lower tropospheric levels (0-10 km) the model grid is set to vertical resolutions of 0.1 km. To save computational time the grid setting is changed to coarser resolutions 10 at higher altitudes (see Table 1).

Time-expensive line-by-line calculations of spectral molecular absorption in the short-wave and long-wave spectral ranges are avoided by employing the sufficiently accurate correlated k-distribution absorption band parametrizations (Kato et al., 1999; Fu and Liou, 1992). Irradiances are then calculated by integrating over the respective parametrized absorption bands and height resolved diurnally averaged heating rates in the short-wave and long-wave spectra are derived from the difference in 15 calculated radiation flux at the particular height intervals solving,

$$\frac{\delta T}{\delta t} = -\frac{1}{c_p \rho(z)} \frac{\delta F_{net}}{\delta z}(z) \tag{1}$$

at any vertical level z. Here, $c_p$ is the specific heat capacity of air at constant pressure, $\rho(z)$ is the altitude dependent air density and $F_{net}(z)/\delta z$ represents the vertical change in net radiative flux at altitude z. The model temperature is parametrized using colocated dropsonde measurements. Reference profiles described by Anderson et al. (1986) are used to parametrize the trace 20 gas concentrations from 0-120 km altitude. However, water vapor profiles and any information on the atmospheric aerosol composition underneath the aircraft are taken from WALES lidar measurements which are interpolated accordingly to fit the model grid.

To minimize uncertainties in surface albedo (Claquin et al., 1998; Liao and Seinfeld, 1998) the bidirectional reflectance distribution function (BRDF; Cox and Munk, 1954a, b; Bellouin et al., 2004) is used. The BRDF derives sea surface albedo





from 10 m-wind speeds measured by dropsondes and sea swell. Based on measurements by the MODIS-Aqua/Terra satellite during the field campaign, sea surface temperature is set to a fixed value of 302 K.

## 2.4 Aerosol optical properties from lidar measurements

In this study, the characterization of aerosol and water vapor profiles in libRadtran is performed using WALES DIAL and depolarization lidar measurements. Therefore, a method to identify profile-regions of different aerosol species and aerosol concentrations using lidar measurements of $\alpha_{p(532)}$, $R_{532}$, $\delta_{p(532)}$ and $r_m$ was developed and is discussed in the following.

### 2.4.1 Aerosol classification

WALES lidar profiles of partilce linear depolarization ratio $\delta_{p(532)}$ can be used to detect and identify Saharan dust marine aerosols in vertical atmospheric columns (Burton et al., 2012; Groß et al., 2013). $\delta_{p(532)}$ for Saharan dust near source regions fluctuates around 0.3 (Freudenthaler et al., 2009; Tesche et al., 2009b; Groß et al., 2011b) and recent studies showed that this ratio remains unchanged after long-range transport across the subtropical North Atlantic Ocean (Burton et al., 2012; Groß et al., 2015). Marine aerosol is composed of sea salt and water-soluble parts and is weakly depolarizing in a moist environment. Dry and stronger depolarizing marine aerosol ($\delta_{p(532)} > 0.04$; Murayama et al., 1999; Sakai et al., 2010) is therefore not expected since relative humidity inside the MBL was found to be always greater than 80 %. As a result, $\delta_{p(532)}$ is a good proxy for the differentiation of mineral dust and less depolarizing marine aerosol (Sakai et al., 2010; Burton et al., 2012; Groß et al., 2013) in NARVAL-II WALES lidar profiles. In this way three aerosol regimes can be determined in the dataset:

I. **pure mineral dust regime**: $\delta_{p(532)} \geq 0.26$,

II. **pure marine aerosol regime**: $\delta_{p(532)} \leq 0.04$,

III. **mixed regime** - marine aerosol mixed with mineral dust: $0.04 < \delta_{p(532)} < 0.26$.

Clear and aerosol-free regions are detected via filtering for no evident particle-backscattering ($R_{532} < 1.2$). Cross sections of an aerosol mask along the HALO flight tracks for libRadtran aerosol input are generated using these criteria.

In addition to the application of the detection scheme, the Saharan origin of detected mineral dust layers is verified utilizing the HYbrid Single Particle Lagrangian Integrated Trajectory model (HYSPLIT; Stein et al., 2015) with NCEP GDAS (National Centers for Environmental Prediction Global Data Assimilation System) input data (shown in Gutleben et al., 2019b). Starting locations and times for the backward trajectory calculations were chosen to match the center of detected mineral dust layers in lidar data.

### 2.4.2 Conversion of aerosol extinction coefficients to aerosol mass concentrations

To run UVSPEC with aerosol input, particle mass concentrations in the classified aerosol regimes have to be determined.

According to Groß et al. (2016), conversion factors can be used to convert measured $\alpha_{p(532)}$ to mineral dust and marine aerosol concentrations per unit volume ($c_{v,dust}, c_{v,marine}$). These factors are taken from results of the AERONET inversion





algorithm by Mamouri and Ansmann (2016) which derived a factor $\nu_{\mathrm{dust}(532)} = c_{\mathrm{v,dust}}/\alpha_{\mathrm{p}(532),\mathrm{dust}} = 6.5 \pm 1.8 \times 10^{-7}\,\mathrm{m}$ for long-range-transported Saharan dust. Due to the similar size distribution of mineral dust and marine aerosols, the AERONET inversion deduced a similar conversion factor of $\nu_{\mathrm{marine}(532)} = c_{\mathrm{v,marine}}/\alpha_{\mathrm{p}(532),\mathrm{marine}} = 7.2 \pm 3.7 \times 10^{-7}\,\mathrm{m}$ for marine aerosol. These values are adopted for this study. Mineral dust and marine aerosol mass concentrations ($c_{\mathrm{m,dust}}, c_{\mathrm{m,marine}}$) are then cal-

culated by multiplying the derived aerosol volume concentrations with typical particle densities of $\rho_{\mathrm{dust}} = 2.5\,\mathrm{g\,cm}^{-3}$ for long-range-transported mineral dust and $\rho_{\mathrm{marine}} = 2.2\,\mathrm{g\,cm}^{-3}$ for marine aerosol. Those particle densities are based on a study by Kaaden et al. (2009) who showed that SALs consist of a mixture of mineral dust particles together with sulfate particles.

The above equations allow the characterization of aerosol mass concentrations in the pure mineral dust regime (I) and marine aerosol regime (II). However, in mixed regimes (III) which appear frequently over the North Atlantic Ocean when

SAL-mineral dust is settling to lower atmospheric levels, the mineral dust contribution to $\alpha_{\mathrm{p}(532)}$ of the total aerosol mixture has to be determined before the application of the conversion coefficients. The aerosol extinction coefficient at 532 nm of a marine-mineral dust aerosol mixture $\alpha_{\mathrm{p}(532),\mathrm{mix}}$ can be written as,

$$
\begin{aligned}
\alpha_{p(532),mix} &= \alpha_{p(532),dust} + \alpha_{p(532),marine} \\
&= (1-x)\,\alpha_{p(532),mix} + x\alpha_{p(532),mix},
\end{aligned} \tag{2}
$$

with $\alpha_{\mathrm{p}(532),\mathrm{marine}}$ and $\alpha_{\mathrm{p}(532),\mathrm{dust}}$ being the marine aerosol and mineral dust particle extinction coefficient at 532 nm and

$x = \alpha_{p(532),dust}/\alpha_{p(532),mix}$.
Using the known lidar ratios of marine and mineral dust aerosol at 532 nm ($S_{\mathrm{p}(532),\mathrm{marine}} \simeq 18$ and $S_{\mathrm{p}(532),\mathrm{dust}} \simeq 47$; Burton et al., 2012; Groß et al., 2013) and following the methods described in Tesche et al. (2009a) and Groß et al. (2011a) one can calculate the fraction $x$ of dust contributing to the total particle extinction coefficient of the mixture,

$$
x = \frac{D_{marine}}{D_{marine} + D_{dust}} \tag{3}
$$

with the coefficients $D_{dust}$ and $D_{marine}$:

$$
D_{dust} := \frac{\delta_{p(532),dust} - \delta_{p(532),mix}}{S_{p(532),dust}(1 + \delta_{p(532),dust})} \tag{4}
$$

$$
D_{marine} := \frac{\delta_{p(532),mix} - \delta_{p(532),marine}}{S_{p(532),marine}(1 + \delta_{p(532),marine})} \tag{5}
$$

Finally, Eq. (3), $\nu_{\mathrm{dust}(532)}$ and $\nu_{\mathrm{marine}(532)}$ as well as $\rho_{\mathrm{dust}}$ and $\rho_{\mathrm{marine}}$ are used to calculate mineral dust and marine aerosol

particle mass concentrations in mixed aerosol regimes (III).

### 2.4.3 OPAC aerosol micro-physical properties

In a last step, converted aerosol mass concentrations are related to aerosol optical properties that are needed for radiative transfer calculations, i.e. the phase function $P(\Theta)$ and the single scattering albedo $\omega$. Those properties are commonly modeled





**Table 2.** OPAC particle type composition of lidar derived aerosol regimes used in the radiative transfer simulations.

| Regime | OPAC component | Mix. ratio |
|---|---|---|
| marine aerosol | sea salt (acc. mode) | 92.8 % |
| | water-soluble | 5.8 % |
| | sea salt (coarse mode) | 1.4 % |
| mineral dust | mineral (acc. mode) | 74.7 % |
| | mineral (coarse mode) | 20.2 % |
| | mineral (nuc. mode) | 3.3 % |
| | water-soluble | 1.8 % |

using size distributions and spectral refrective indices of the desired aerosol species. Model-assumptions always represent some source of uncertainty. For example, Yi et al. (2011) showed that different representations of particle shapes result in a change of $P(\Theta)$ and can cause up to 30 % difference in the dust radiative forcing at top of the atmosphere (TOA). To minimize errors resulting from wavelength-interpolations Hess et al. (1998) established the readily available spectrally resolved OPAC

database (Optical properties of Aerosols and Clouds) which includes modeled information on the above mentioned aerosol optical properties for 61 wavelengths in the spectral range from 0.25-40 $\mu$m for various aerosol species. OPAC is a widely used data base in aerosol models and retrievals (e.g. Kim et al., 2004; Liu et al., 2004; Patadia et al., 2009) as well as general circulation and climate models for calculations in both the short-wave and the long-wave spectra. Thus, it is an appropriate tool to link lidar derived aerosol mass concentrations to aerosol optical properties in the classified aerosol regimes.

OPAC sea salt and water-soluble particle microphysical properties are modeled under the assumption of spherical particles using Mie-Theory (Mie, 1908). The assumption of spherical particles is legitimate for radiative transfer simulations in the period of NARVAL-II since no dry and aspherical marine aerosol particles are expected at observed relative humidities of greater 80 % inside the derived marine aerosol regimes (Murayama et al., 1999; Sakai et al., 2010). Thus, a humidity dependent marine aerosol composition which refers to WALES measurements of water vapor mixing ratios together with dropsonde-

derived temperature profiles is used in this study (see Table 2).

    Mineral dust particles however, are characterized by highly irregular shapes (Falkovich et al., 2001; Kandler et al., 2011). Hence, an assumption of spherical mineral dust particles in radiative transfer models using Mie-Theory is inappropriate and may lead to wrong results. Especially if particles are significantly larger than the transmitted wavelength (which is the case for most backscatter lidar systems) phase functions of aspherical particles are characterized by an increased amount of sideward

but a reduced amount of backward scattering compared to spherical particles (Koepke and Hess, 1988; Gobbi et al., 2002; Nousiainen, 2009; Wiegner et al., 2009; Gasteiger et al., 2011; Sakai et al., 2014). For this reason mineral dust particle microphysical properties were updated in the latest OPAC version (v4.0; Gasteiger et al., 2011; Koepke et al., 2015) and are now calculated by means of the T-matrix method (Waterman, 1971) under assumptions of an aspect ratio distribution for prolate spheroids observed during the Saharan mineral dust experiments SAMUM-I and SAMUM-II (Kandler et al., 2009, 2011).



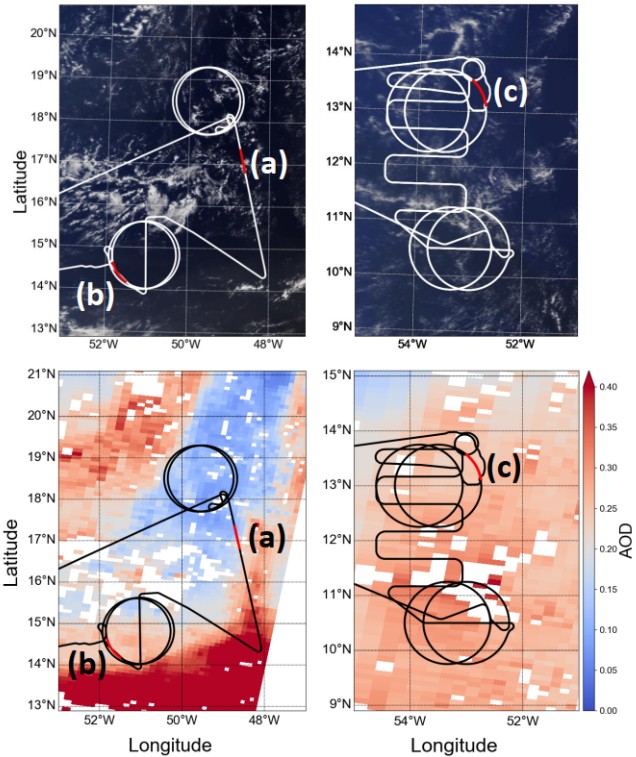

**Figure 1.** Flight tracks of the research flights conducted on 12 (right) and 19 (left) August 2016 on top of images showing MODIS (MODerate-resolution Imaging Spectroradiometer) true color (top) and Aerosol Optical Depth (AOD, bottom) around 13:40 UTC. Red lines and labels indicate the discussed flight segments (cases (a), (b) and (c)).

Several studies have shown that T-matrix theory substantially improves the agreement between measured and modeled aerosol optical properties of aspherical mineral dust particles (Mishchenko et al., 1997; Kahnert et al., 2005; Gasteiger et al., 2011) and are thus motivating its use in this study.

Results obtained from measurements during SALTRACE (Weinzierl et al., 2017) showed that the size distribution of long-
5   range-transported mineral dust does not change significantly compared to the distributions measured at source regions. Gravitational settling processes of large sub-micron particles in the course of the SAL-transatlantic transport are of a smaller magnitude than expected from Stokes gravitational settling calculations. Moreover, Denjean et al. (2015) found that the chemical composition and hygroscopy of mineral dust remains unchanged after long-range transport. Thus, a mixture proposed by Hess et al. (1998) which consists of four OPAC v4.0-components for desert aerosol optical properties is assumed in this paper (see
10   Table 2).

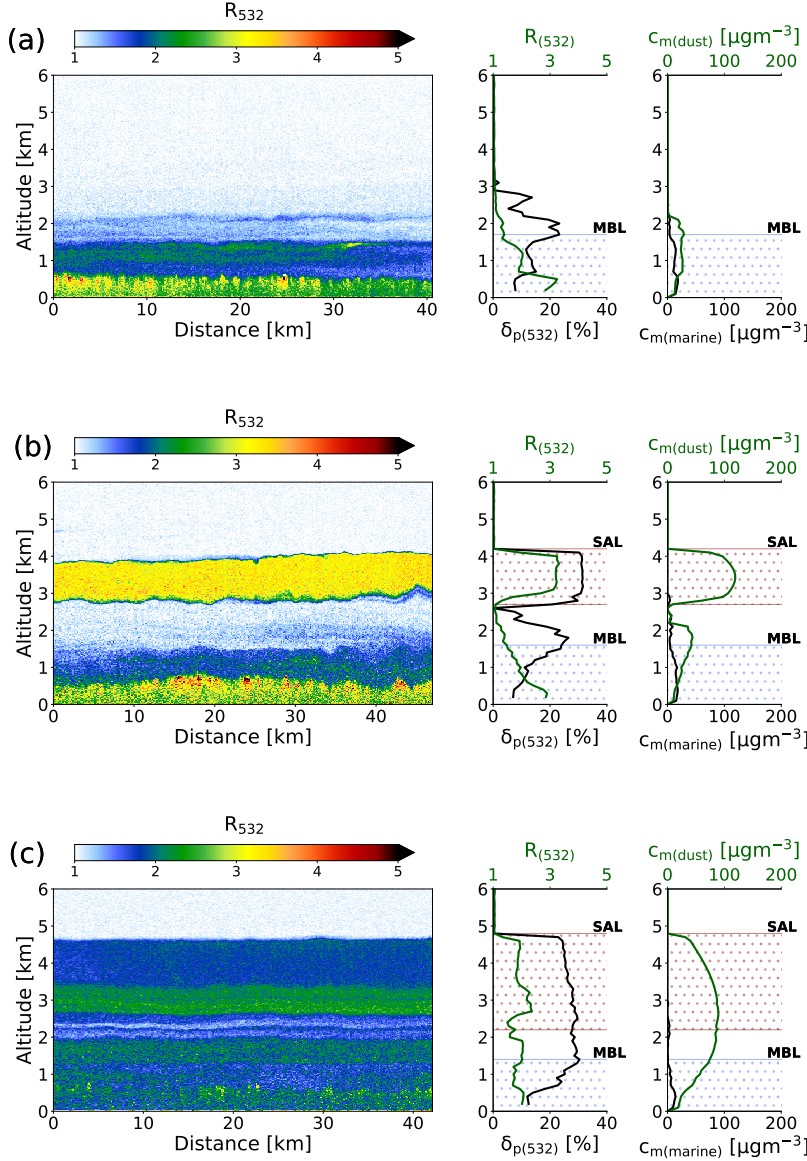

**Figure 2.** Left: cross-sections of WALES backscatter ratio $R_{(532)}$; Middle: averaged profiles of particle linear depolarization ratios $\delta_{p(532)}$ and backscatter ratio $R_{(532)}$; Right: derived aerosol mass concentrations ($c_{m(dust)}$ (red) and $c_{m(marine)}$ (blue)) for case (a) and (b) on 19 Aug 2016 as well as case (c) on 12 Aug 2016.





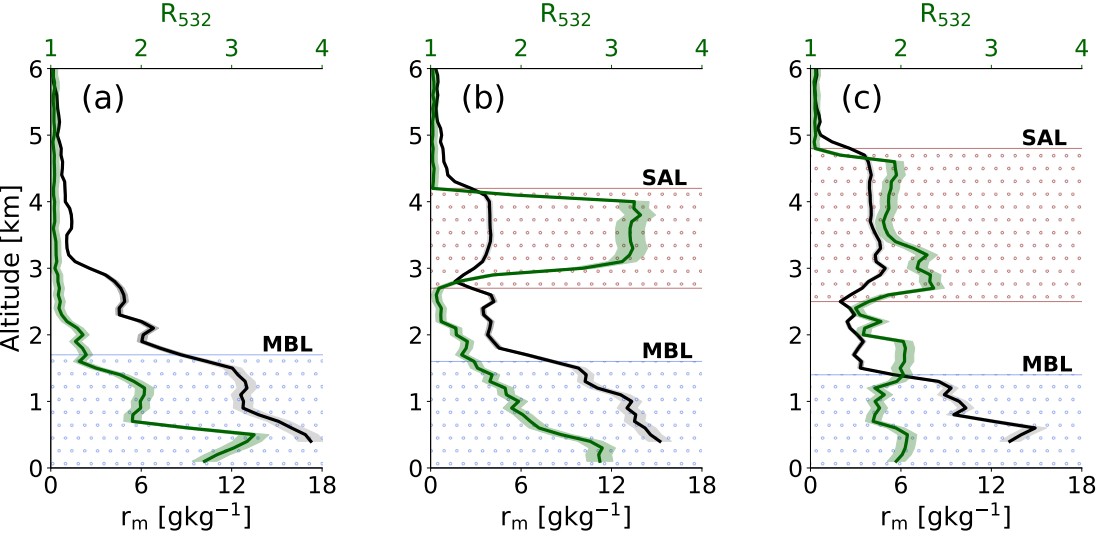

**Figure 3.** Averaged vertical profiles of backscatter ratio ($R_{532}$) and water vapor mass mixing ratio ($r_m$) for cases (a),(b) and (c). Shaded: estimated measurement uncertainties. Colored height intervals highlight the Saharan air layer (SAL; orange/dotted) and marine boundary layer (MBL; light-green/dotted).

## 3 Results

### 3.1 NARVAL-II lidar measurements

During two NARVAL-II-research flights on 12 and 19 August 2016 pronounced elevated SALs could be observed. On 12 August (takeoff: 11:43 UTC; landing: 19:37 UTC) dust layers covered the whole research area. This is also seen in MODIS

5  (Moderate Resolution Imaging Spectroradiometer) total column Aerosol Optical Depth (AOD) which took values around 0.3 along the whole flight track (Figure 1). In contrast, the research flight on 19 August (takeoff: 12:28 UTC; landing: 20:52 UTC) led over trade wind regions comprising elevated mineral dust layers (AOD > 0.3) as well as regions free of mineral dust and is therefore suitable for a comparison of radiative effects in SAL-influenced regions and SAL-free regions.

In this study three representative cloudless 5 min-lidar cross sections observed during these two research flights are used

10  for radiative transfer calculations (Figures 2). While the first case represents a SAL-free scenario with some residual mineral dust at low atmospheric levels, the other two cases are both characterized by a long-range-transported SAL. Measured vertical profiles of both $R_{532}$ and $\delta_{p(532)}$ are used for the detection of the SAL-outlines. From enhanced values of $R_{532}$ and typical values of $\delta_{p(532)}$ for mineral dust the vertical extent of SALs can be determined. All the three selected cross sections are of approximately 50 km length and described in the following:





(a) *SAL-free case - 19 Aug 2016 (16:51-16:56 UTC):*

This case represents a SAL-free measurement segment during NARVAL-II. Lidar profiles of $R_{532}$ and $\delta_{p(532)}$ show no dust-signatures in altitudes greater 3 km. In the MBL both marine and mineral dust aerosol particles coexist (settling dust particles) and a maximum aerosol extinction coefficient of $0.03\,\mathrm{km^{-1}}$ was measured inside the MBL. $\delta_{p(532)}$ ranges from roughly 0.05-0.25 and aerosol mass concentrations of both marine and mineral dust aerosol reach maximum values of approximately $30\,\mu\mathrm{g\,m^{-3}}$.

(b) *Elevated SAL - 19 Aug 2016 (14:26 - 14:31 UTC):*

The second case represents a scenario with a detected elevated and long-range-transported SAL extending from 3-4 km altitude. The SAL shows increased backscatter ratios around 3.5 and high particle linear depolarization ratios >0.26. $\delta_{p(532)}$ and $R_{(532)}$ profiles feature sharp gradients to the above free-troposphere and to lower atmospheric levels. The SAL itself is associated with evenly distributed mineral dust mass concentrations of approximately $120\,\mu\mathrm{g\,m^{-3}}$ and aerosol extinction coefficients around $0.07\,\mathrm{km^{-1}}$. Marine aerosols are mainly confined to the MBL which extends from 0.0-1.6 km altitude. Similar to case (a) the aerosol composition within the MBL is not exclusively characterized by marine aerosols but also contains portions of mineral dust aerosols. An intermediate layer showing small values of $\delta_{p(532)}$, $c_{m(dust)}$ and $R_{(532)}$ is located in-between the MBL and the SAL and ranges from 1.6-2.8 km altitude.

(c) *Thick SAL - 12 Aug 2016 (14:26 - 14:31 UTC):*

The third scenario represents the geometrically thickest SAL that has been observed during the whole NARVAL-II campaign. Measured $\delta_{p(532)}$ around 0.3 clearly indicate the presence of Saharan mineral dust from ground level to almost 5 km altitude. The $R_{(532)}$-profile also shows enhanced particle backscatter greater 2.0 in these altitudes. Profiles of aerosol mass concentration highlight a pure dust regime ($c_{m(dust)} \approx 100\,\mu\mathrm{g\,m^{-3}}$; aerosol extinction coefficients around $0.06\,\mathrm{km^{-1}}$) at altitudes ranging from approximately 1.5-5.0 km altitude transitioning to a mixed marine and dust aerosol regime at lower atmospheric levels (0-1.5 km). The lidar measurements again do not indicate a pure marine aerosol regime at low altitudes. Both marine and settling mineral dust aerosol is found in the MBL (0-1.5 km).

All observed SALs during the NARVAL-II period were associated with enhanced concentrations of water vapor compared to the surrounding dry free trade-wind environment (Gutleben et al., 2019a). $r_m$ and $R_{(532)}$ in the lidar profiles (b) and (c) also show a distinct correlation (Figure 3). The SALs show almost uniformly increased water vapor mixing ratios ranging from $3$-$5\,\mathrm{g\,kg^{-1}}$ compared to the surrounding free-troposphere. (case (b): 2.8-4.2 km; case (c): 2.5-4.8 km). Case (a) however, indicates that no distinct correlation of enhanced $r_m$ and $R_{(532)}$ could be observed in a SAL-free troposphere. The profile shows a drop of $r_m$ to values smaller $1\,\mathrm{g\,kg^{-1}}$ at altitudes greater 3 km, indicating the transition from the MBL to the dry free troposphere. Such a drop in humidity (which is coming along with a strong trade wind inversion caused by Headly cell subsidence) was observed during most SAL-free periods in NARVAL-II, and is discussed by Gutleben et al. (2019b) in the framework of a detailed dropsonde analysis.





Enhanced water vapor concentrations in SAL-altitudes are already seen at the beginning of the transport (using HYSPLIT backward trajectories), when analyzing radiosonde profiles from operationally launched sondes at Dakar/Senegal four days before the measurements on 8 Aug 2016 and 15 Aug 2016 (Gutleben et al., 2019a).

### 3.2 Saharan Air Layer Heating rates

5 Profiles of calculated short-wave, long-wave and net heating rates (24 h-averaged) for the three selected case studies are shown in Figure 4. Since WALES is able to measure both water vapor mixing ratios and aerosol optical properties, total heating rate profiles and contributions of mineral dust to total heating rate profiles can be derived. The dust-contribution to the total heating rate is derived as the difference between heating rates that consider dust in the model and heating rates with no dust in the model atmosphere.

10 Observed SALs in case (b) and (c) are well mixed (constant potential temperature $\Theta$ around 315 K) and show enhanced water vapor mass mixing ratios in the range from 2-5 $\mathrm{g\,kg^{-1}}$ compared to the surrounding dry free atmosphere. Both profiles have strong gradients of $r_m$ and $\Theta$ at the upper edge of the SAL (at the boundary to the above dry and aerosol-free trade wind atmosphere) indicating the two well-known SAL-related bounding inversions (Lilly, 1968). The MBL in all three cases is characterized by high relative humidities ($r_m$: 10-16 $\mathrm{g\,kg^{-1}}$) and is capped by a temperature inversion (trade wind inversion) 15 and a pronounced hydrolapse ($r_m$ drops from >15 to approximately 5 $\mathrm{g\,kg^{-1}}$).

Calculated profiles of diurnally averaged mineral dust short-wave heating rates for mean profiles of case (b) and (c) indicate an atmospheric heating of less than 0.5 $\mathrm{K\,d^{-1}}$ in SAL-altitudes. Maximum short-wave heating is hereby found in altitudes of highest dust mass concentration (case (b): ~3.5 km; case (c): 2.5 km). Also long-wave cooling rates of dust are strongest at altitudes of highest dust mass concentration (~0.2 $\mathrm{K\,d^{-1}}$). This results in a small net warming effect of long-range-transported 20 dust aerosols of approximately 0.3 $\mathrm{K\,d^{-1}}$ for both cases (b) and (c). The net mineral dust radiative heating rate for the SAL-free case (a) is limited to the lowest atmospheric levels and takes small values of less than 0.1 $\mathrm{K\,d^{-1}}$.

Due to water vapor absorption and emission the total heating and cooling rate profiles have a completely different shape. Largest water vapor absorption of solar radiation takes place at the uppermost levels of the SAL leading to strong heating at these levels. Long-wave cooling due to emission of radiation towards space is also strongest at the top edge of the SAL since 25 there is no heating from atmospheric counter radiation from higher atmospheric levels. This is why greatest total heating and cooling rates are found at the upper edge of both observed SALs (short-wave: ~2.2 $\mathrm{K\,d^{-1}}$(both cases); long-wave: -6 $\mathrm{K\,d^{-1}}$ (case (b)) and -7 $\mathrm{K\,d^{-1}}$ (case (c)).

Adding short-wave and long-wave heating rates results in SAL-net heating rates that are entirely negative for both cases, taking values from -1.0 to -3.5 $\mathrm{K\,d^{-1}}$ (case (b)) and -0.5 to -5.5 $\mathrm{K\,d^{-1}}$ (case (c)). Moreover, an intensification of net radiative 30 cooling with height towards the top of the SAL is evident.

Another increase in short-wave heating and long-wave cooling rates is found within the MBL which is characterized by a moist mixture of mineral dust and marine aerosols in all three cases. However, the shape of the MBL heating rate profile in SAL-influenced regions differs to those in SAL-free regions. While for the SAL-free scenario strong cooling at the top of the MBL can be observed (strongly pronounced MBL-inversion; case (a)), SAL-influenced regions show less cooling (weakly

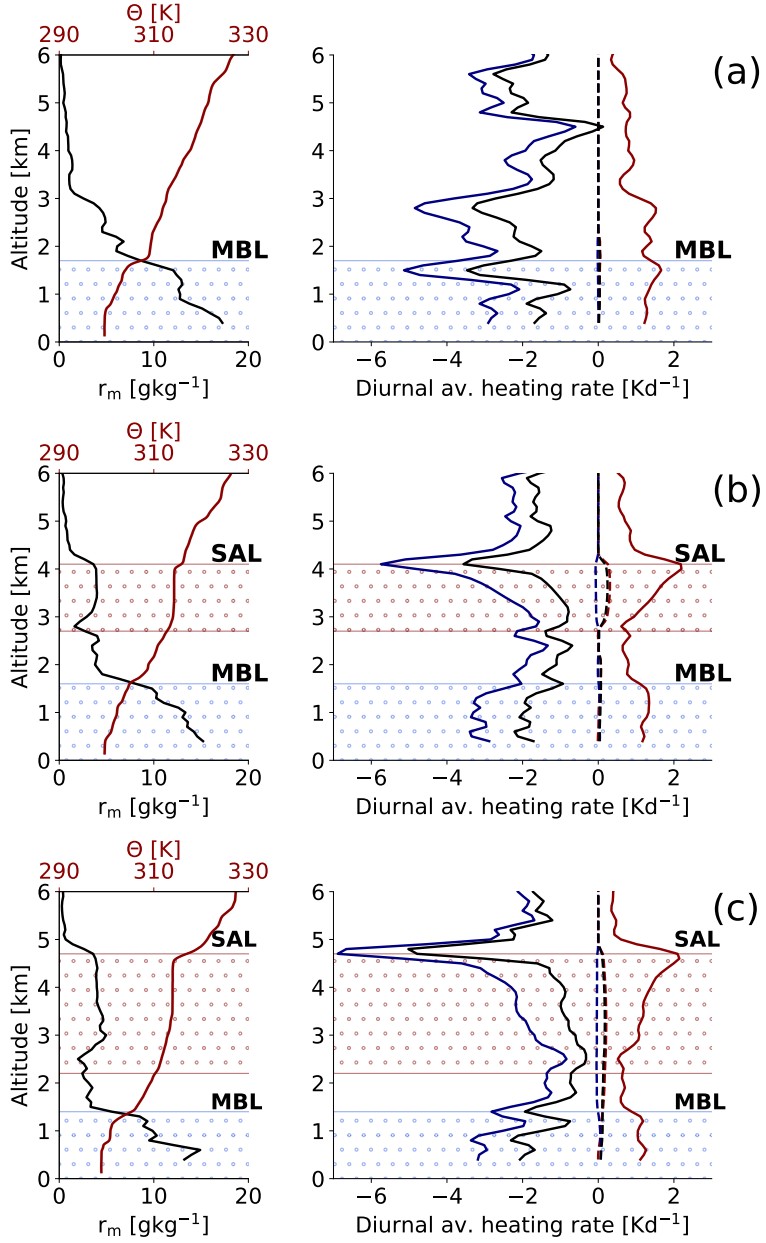

**Figure 4.** Left: the vertical profiles of derived water vapor mass mixing ratio from DIAL measurements (blue) and potential temperature from dropsonde measurements (red) for the three scenarios. Right: the diurnally averaged net (black), short-wave (red) and long-wave (blue) heating rates (solid). Dashed lines illustrate the aerosol-contributions to the total heating rate profiles. Colored height intervals highlight observed Saharan air layers (SAL; red-dotted) and marine boundary layers (MBL; blue-dotted).





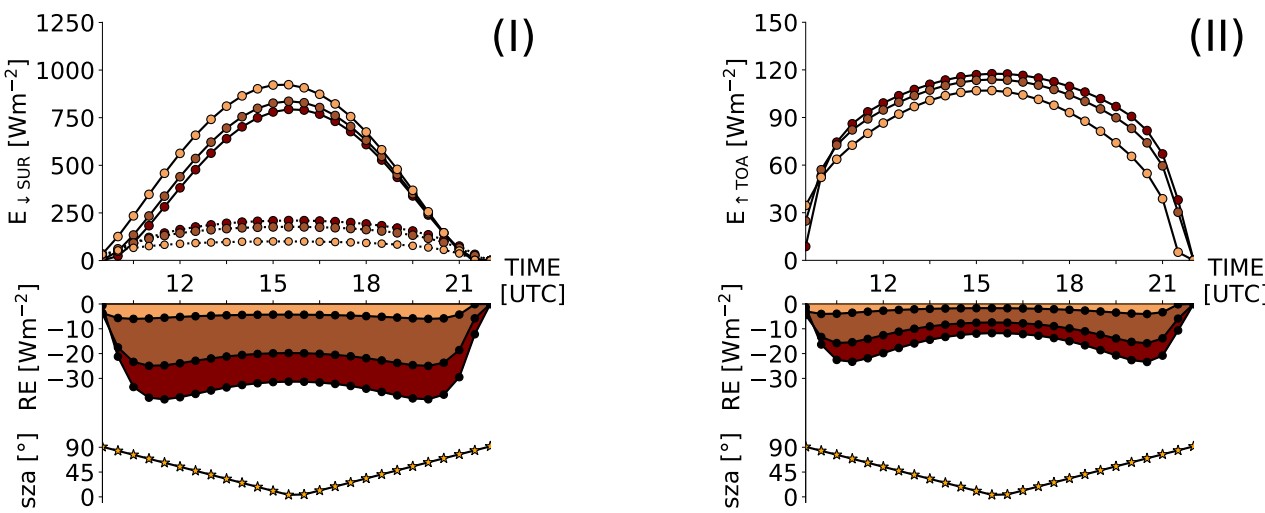

**Figure 5.** Top: Diurnal cycles of the modeled down-welling short-wave irradiances at surface level (I - $E_{\downarrow SUR}$ : direct (solid), diffuse (dashed)) and up-welling short-wave irradiances at top of the atmosphere (II - $E_{\uparrow TOA}$). Due to the different measurement locations and small differences in corresponding solar zenith angles (sza) calculated curves are slightly shifted in x-direction. Bottom: The corresponding diurnal cycles of the modeled Saharan dust short-wave radiative effects (RE) at surface level (I) and top of the atmosphere (II). A representative sza for the whole measurement region is shown at the bottom ($11°N$ and $55°W$). Colors indicate the three cases: light-orange (case (a)), orange (case (b)), dark-orange (case (c)).

pronounced MBL-inversion; cases (b) and (c)). Short-wave, long-wave as well as net heating rate profiles calculated for the dust-free case (a) show no distinct features above the MBL. In this case peak values of atmospheric heating and cooling correlate with regions of strongest gradients in $r_m$ (maximum long-wave cooling: $-5\,\mathrm{K\,d^{-1}}$; maximum short-wave heating: $1.8\,\mathrm{K\,d^{-1}}$). This emphasizes the dominating effect of water vapor on atmospheric heating.

## 3.3 SAL radiative effects at surface level and top of the atmosphere

Saharan dust short-wave radiative effects at surface level and TOA (Figure 5) are investigated by analyzing modeled solar zenith angle dependent short-wave irradiances for the three discussed scenarios. It is assumed that the observed profiles do not change and remain stationary within a 24 h time frame. Saharan dust short-wave radiative effects at surface level ($RE_{SUR}$) and top of the atmosphere ($RE_{TOA}$) are inferred as the difference between modeled irradiances considering mineral dust particles in the model atmosphere ($E_{\downarrow tot(SUR)}$, $E_{\uparrow tot(TOA)}$) and irradiances calculated under assumption of no dust aerosol in the atmosphere ($E_{\downarrow nodust(SUR)}$, $E_{\uparrow nodust(TOA)}$),

$$RE_{SUR} = E_{\downarrow tot(SUR)} - E_{\downarrow nodust(SUR)}. \tag{6}$$





and,

$$\mathrm{RE_{TOA}} = -(\mathrm{E_{\uparrow tot(TOA)}} - \mathrm{E_{\uparrow nodust(TOA)}}), \tag{7}$$

Downward and upward irradiances are primarily determined by solar elevation, therefore having a symmetrical shape with maxima at noon (around 15:30 UTC). The longer the slant path of solar rays through SALs, the more Mie- and Rayleigh-scattering processes and the larger the fraction of backscattered light to space. As a consequence, $\mathrm{RE_{SUR}}$ and $\mathrm{RE_{TOA}}$ show minimum magnitudes at low solar zenith angles around noon and maximum magnitudes at intermediate zenith angles in the morning and evening hours.

For case (c) a maximum negative $\mathrm{RE_{SUR}}$ of almost -40 W m$^{-2}$ is calculated for morning and evening hours (11:30 and 20:00 UTC). At noon the diffuse fraction of down-welling irradiance ($\mathrm{E_{\downarrow diff}}$) at surface level reaches 200 W m$^{-2}$ making one third of $\mathrm{E_{\downarrow tot}}$ at surface levels. Calculations of $\mathrm{RE_{TOA}}$ show similar results for case (c). At intermediate zenith angles the diurnal maximum in atmospheric backscattering causes a maximum negative $\mathrm{RE_{TOA}}$ of -25 W m$^{-2}$. $\mathrm{E_{\uparrow tot}}$ at TOA reaches maximum values of nearly 120 W m$^{-2}$ at midday. For case (b) a lidar-derived $\mathrm{RE_{SUR}}$ of -25 W m$^{-2}$ is computed for intermediate zenith angles. It shows a local minimum at noon (-20 W m$^{-2}$). Compared to case (c), $\mathrm{E_{\downarrow diff}}$ at noon is slightly smaller (180 W m$^{-2}$). This reduced diffuse fraction is also evident when looking at $\mathrm{RE_{TOA}}$. Due to the decrease in scattering, $\mathrm{RE_{TOA}}$ reduces to a minimum of -17 W m$^{-2}$ for case (b). Compared to the other cases, case (a) shows the smallest values of $\mathrm{RE_{SUR}}$ (>-8 W m$^{-2}$) with weakly pronounced maxima during sunrise and sunset. $\mathrm{RE_{TOA}}$ is also weakest (>-5 W m$^{-2}$).

## 4   Discussion

In this study the effects of mineral dust particles and water vapor on radiative heating rate profiles in SAL-influenced regions were studied. It was found, that the enhanced water vapor concentrations within the SALs cause a decrease of radiative heating rates towards the top of the SALs. This negative gradient with height is in agreement with results found by Kim et al. (2004) who focused on the effect of enhanced water vapor concentrations on atmospheric heating rates within Asian dust plumes. They also highlighted that derived atmospheric heating rates within the dust plumes are altered by enhanced water vapor concentrations and compared heating rate calculations including measured water vapor profiles to reference profiles. Calculated maximum short-wave heating and long-wave cooling rates were also found to be shifted from the center to the top of the dust layer when including the measured water vapor profile in their calculations. The strong negative trend of the heating rate profiles within the SALs is supposed to decrease the static stability in the layers and to promote vertical mixing and convective development. Vertical mixing in the SALs during their transport over the Atlantic Ocean was already proposed by Gasteiger et al. (2017) in an integrated study of active remote sensing, in-situ measurements and optical modeling. They suggested that vertical mixing within the SALs may counteract the Stokes gravitational settling during transport. Dropsonde measurements discussed in Section 3.1 confirm the well mixed and neutrally stratified layering in the interior of the SALs ($\Theta = \mathrm{constant}$). It is bounded by inversions at the top and the bottom of the SALs (increase of net heating rates and positive changes of potential temperature), which separate the turbulent interior from the less turbulent free troposphere. An early model analysis by Lilly (1968) already suggested that these inversions are caused by radiative heating in SAL-altitudes.





Enhanced moisture inside the SAL also has an impact on the cooling and heating rate profile of the MBL. The dust-laden cases are found to show less infrared cooling at the top of the MBL compared to the SAL-free case. The elevated and moist SAL is associated with down-welling long-wave irradiance which counteracts radiative cooling inside the MBL, thus weakening the cooling of the MBL. This leads to an almost homogeneous heating rate profile within the MBL. The SAL-free case in contrast

indicates a negative heating rate gradient with strong negative heating rates at the top of the MBL. The nearly constant heating rate profile in the MBL counteracts convective development. Stevens et al. (2017) used idealized distributions of water vapor in the lower atmosphere to highlight the impact of elevated moist layers on the vertical distribution of heating rates. They claimed that such layers reduce the cooling at lower atmospheric levels and therefore modify the state of the boundary layer potentially inducing low-level circulations. A theoretical study by Naumann et al. (2017) already explained that variations in

infrared cooling due to vertical gradients of tropospheric moisture may drive atmospheric circulations in trade wind regions. A correlation between long-range-transported elevated SAL and subjacent low-level cloudiness has been observed by Gutleben et al. (2019b). Our analysis showed reduced cloud fractions and lower cloud top heights in trade wind regions comprising elevated mineral dust layers compared to dust-free regions. Moreover, observed dusty regions contained smaller clouds and showed greater cloud gaps than dust-laden regions.

Kanitz et al. (2013) studied the Saharan dust radiative effect at surface and TOA near source regions in the vicinity of the Cap Verde islands using shipborne aerosol Raman/polarization lidar measurements to parametrize the atmospheres aerosol composition. Based on their lidar measurements on 1 May 2010 they deduced a maximum diurnal dust short-wave radiative effect of approximately -60 $\mathrm{W\,m^{-2}}$ at surface level and -42 $\mathrm{W\,m^{-2}}$ at TOA. These results are in good agreement with calculated radiative effects in this study. However, results in this paper are of a slightly smaller magnitude due to thinning of the SAL

during long-range transport (-40 $\mathrm{W\,m^{-2}}$ at the surface and -25 $\mathrm{W\,m^{-2}}$ at TOA). (Foltz and McPhaden, 2008) found that less down-welling solar radiation in dust-laden regions may cause gradients in sea surface temperature and potentially impacts the evolution of clouds in the MBL.

During NARVAL-II, the majority of mineral dust particles was always located above the MBL and inside the SAL. However, during previous field campaigns it was observed that the vertical distribution of long-range-transported mineral dust can be

highly variable (Reid, 2002). During the Puerto Rico Dust Experiment (PRIDE) in summer 2000, for example, the majority of dust was in some cases observed to be located at lower atmospheric levels inside the MBL and in other cases it was observed to be located inside the SAL. A distinct seasonal pattern of Saharan dust transport towards the Atlantic Ocean was already found by Chiapello et al. (1995), who studied the vertical distribution of mineral dust particles at the beginning of long-range transport at the Cap Verde islands. They found that in contrast to the summer months, wintertime dust-transport towards the

Atlantic Ocean is mainly taking place at lower atmospheric levels. Questions regarding the reasons for these variable vertical distributions of mineral dust and whether there is a certain seasonal pattern in the vertical distribution not only at the beginning but also throughout the long-range dust transport can hopefully be answered in near future by analyzing data collected during the recent EUREC[4]A field campaign (ElUcidating the RolE of Clouds-Circulation Coupling in ClimAte; Bony et al., 2017) in January 2020 .



## 5 Summary and conclusions

This study investigated the effects of dust and water vapor in long-range-transported SALs on atmospheric heating rates and radiative transport on the basis of airborne lidar measurements over the western subtropical North Atlantic Ocean. Simultaneously measured profiles of water vapor mass mixing ratios and aerosol optical properties were used to characterize both the
vertically resolved aerosol and water vapor composition in radiative transfer simulations.

    Lidar measurements in Saharan dust-laden regions indicated enhanced concentrations of water vapor in SAL-altitudes and radiative transfer simulations revealed that water vapor plays the dominant role for atmospheric heating rates in these heights. Compared to water vapor, dust aerosol was identified to have a minor effect on total heating rates in SAL-altitudes showing small positive maximum heating rates of 0.3-0.5 $\mathrm{K\,d^{-1}}$ in the short-wave and slightly negative maximum cooling rates of -0.1
to -0.2 $\mathrm{K\,d^{-1}}$ in the long-wave spectrum at altitudes of highest aerosol mass concentration. Water vapor, however, was found to contribute much stronger to total SAL-heating rates with maximum short-wave and long-wave heating of 1.8-2.2 $\mathrm{K\,d^{-1}}$ and -6 $\mathrm{K\,d^{-1}}$ to -7 $\mathrm{K\,d^{-1}}$ at the uppermost levels of the SAL. As a result, calculated net heating rates inside SALs are entirely negative and decrease with altitude.

    SALs were also found to have a possible impact on cloud development in the MBL. Besides possible impacts on low-level
circulations, SALs introduce additional atmospheric counter-radiation towards the top of the MBL. As a result, MBL tops in dust-laden regions do not experience as much cooling as in SAL-free regions. This is also indicated by the heating rate profile in SAL-regions which is increasing with altitude and therefore counteracts the development of convection.

    Last but not least, NARVAL-II lidar data were used to quantify the radiative effect of long-range-transported Saharan dust at surface level and top of the atmosphere. Maximum short-wave radiative effects of -40 $\mathrm{W\,m^{-2}}$ (surface) and 25 $\mathrm{W\,m^{-2}}$ (TOA)
were found at intermediate zenith angles.

    Summed up, radiative transfer calculations with NARVAL-II lidar data input highlighted the importance of correct representations of water vapor profiles in radiative transfer models and depicted the influence of mineral dust on the modification of solar irradiance throughout the atmosphere.

*Data availability.* The data used in this publication was collected during the NARVAL-II (Next-generation Aircraft Remote-Sensing for
Validation Studies-II) campaign and is made available through the DLR Institute for Atmospheric Physics in the HALO database (German Aerospace Center, 2016; doi: 10.17616/R39Q0T).

*Author contributions.* In the framework of the NARVAL-II field experiment MW and SG contributed to carry out all airborne lidar measurements used in this study. MW did the initial data processing. MG performed the analytic computations, analyzed the dataset and performed radiative transfer calculations with help BM and under supervision of SG. MG and SG took the lead in writing the manuscript. All authors
discussed the results and contributed to the final manuscript.





*Competing interests.* The authors declare that they have no conflict of interest.

*Acknowledgements.* The authors like to thank the staff members of the DLR HALO aircraft from DLR Flight Experiments for preparing and performing the measurement flights. NARVAL-II was funded with support of the Max Planck Society, the German Research Foundation (DFG Priority Program: HALO-SSP 1294) and the German Aerospace Center (DLR). This study was funded by a DLR VO-R young

5  investigator group within the Institute of Atmospheric Physics.



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
