# Peer review of "Radiative effects of long-range-transported Saharan air layers as determined from airborne lidar measurements"

_Atmospheric Chemistry and Physics, 2020_

## Referee Comment (RC1) · Anonymous Referee #2 · 11 Jun 2020

The authors present vertically resolved measurements of aerosol optical properties and water vapor mixing ratios at the same time combining different lidar techniques. The data were collected with an airborne lidar instrument in the long-range transport regime of Saharan dust during the NARVAL-II campaign. Using the profile measurements in the Saharan Air Layer (SAL) as input, they perform radiative transfer calculations. They found that the water vapor inside the SAL is the driving force for the radiative cooling at top of the SAL. The dust particles have only a minor contribution to the net heating rates. The heating rate inside the SAL is decreasing with height to a maximum cooling at the top of the SAL. With these findings the authors gave sound explanation for the vertical mixing within the SAL which counteracts the gravitational

settling of the large dust particles during long-range transport from Africa towards the Caribbean.

The findings presented in the manuscript contribute significantly to the scientific understanding of the long-range transport of the Saharan air layer over the Atlantic Ocean and publication is highly recommended. Although the main message of the manuscript was already given in a publication by the same authors (Gutleben et al., GRL 2019), the current manuscript contributes to a deeper understanding of the process by presenting 3 well-analyzed case studies and the calculation of the radiative effect of the SAL at surface and top of atmosphere. The authors highlight the importance of the correct water vapor profile in radiative transfer simulations in case of the SAL. Important for the quality of the study are the simultaneous measurements of the vertical profiles of aerosol optical properties and water vapor replacing assumptions which are often made by studies relying on passive sensors.

The manuscript is well written and could be published after minor revisions which are addressed in the following.

The main points to be considered are:

I. A clear definition of the term "marine boundary layer" (MBL) is missing. In the presented case studies (Fig. 4) the so-called MBL reaches up to the trade wind inversion at about 1.4 – 1.8 km height. Later, it is stated that the SAL is confined at the bottom by the trade wind inversion which is not in line with the indicated SAL in Fig. 4. Please explain in a more quantitative way how the upper limit of the MBL and the lower limit of the SAL are determined in your study. Various attempts to describe the vertical stratification in the Caribbean are found in literature ("convective marine boundary layer (CMBL)" in Groß et al., 2016, "sub-cloud layer (SCL)" and "intermediate layer (IL)" in Jung et al., 2013, "marine aerosol layer (MAL)" in Rittmeister et al., 2017). The convective part (CMBL or SCL) of the MBL or MAL is clearly visible in Fig. 2 and reaches up to around 600 m height. A definition of the MBL as used in the present study would improve the manuscript.

II. Some comments concerning the aerosol classification (Sec. 2.4.1):

You discuss only the contribution of pure mineral dust and pure marine aerosol and its mixture. What about contributions of other aerosol types like pollution from the African continent? At this time of the year, it is not very likely, but should be mentioned.

The effect of dry marine aerosol on the depolarization ratio was even studied at Barbados and is described in Haarig et al., 2017a.

III. Some comments concerning Section 2.4.2:

The lidar ratios for dust (47 sr) and marine aerosol (18 sr) are based on literature from different locations around the globe. However, Groß et al., 2015, reported higher dust lidar ratios of 56  $\pm$  7 sr for Barbados. Could you derive with your high spectral resolution lidar (HSRL) actual lidar ratios of the pure types during your campaign to judge which lidar ratios are more appropriate for your situation?

What values for the particle linear depolarization ratio have you used in Eq. 4 and 5? 0.26 and 0.04?

From the text it is not clear whether you have calculated the conversion factors (extinction to volume) using the method described in Mamouri and Ansmann (2016) or whether you have taken the values from the literature. Looking into the cited paper, I can't find your conversion factors.

IV. For your radiative transfer calculations, it is important that no cirrus cloud is present above the scene. The upper panel of Fig. 1 (not yet used in the text) supports the absence of cirrus clouds.

Minor comments:

1. Pii, L12 + L26: Using the term "we" for studies by the same authors is a matter of taste.

СЗ

2. Pii, L21: Gutleben et al. (2019x), x = a or b?

3. Pii, L 6: Haarig et al., 2018, describes smoke, you probably mean Haarig et al., 2017b.

4. Piv, Eq (1) What is  $\delta T/\delta t$ ? And two lines later a  $\delta$  is missing ( $\delta F$ ).

5. Pv, L8: Saharan dust \_and\_ marine aerosols

6. Pv, L11: Burton et al., 2012, is not a good reference for the unchanged dust properties. It describes the aerosol classification and gives just a case study showing mineral dust. Look for other references.

7. Pvi, Eq (2) - (5) and the whole page: Indices should not be written in italic.

8. Pvii, Table 2: The percentages for the mixing ratio refer to which quantity (volume, number, ...)?

9. Pix, Fig. 2, caption: The profile are not shown in red and blue, but green and black.

10. Fig. 2 and Pxi, case (a): There is still a thin layer of some depolarizing particles between 1.7 and 2.5 km height. You are right, there are no dust signatures above 3 km height. But what about the particles above the MBL top? Please discuss this issue shortly.

11. Fig. 3, caption: light-green -> light-blue

12. Pxi, L27/28: "Case (a) however, indicates that no distinct correlation of enhanced rm and R(532) could be observed in a SAL-free troposphere." What do you mean by "no distinct correlation"? Both curves are decreasing with height.

13. Pxii, L5: 24 h-averaged – The explanation follows some pages later: "It is assumed that the observed profiles do not change and remain stationary within a 24 h time frame." (Pxiv)

14. Pxii, L11: 2–5 g/kg, one page before you stated 3–5 g/kg

15. Fig. 4: DIAL measurements are shown in black not blue.

16. Fig. 4 right: Black and dark blue are difficult to distinguish. Please take a different color.

17. Sect. 3.3: According to the text, the radiative effects are calculated for the Saharan dust particles disregarding the water vapor. Is this the case? Please state it clearly.

18. Pxv, L8-16: This paragraph should be carefully rewritten. "Calculations of RETOA show similar results for case (c). (L10)" – Similar to what? In case of negative numbers, what is considered a minimum? (L13-15)

19. Pxvi, L20: Reference in brackets.

20. Pxvi, L34: January/February 2020

21. Pxvii, L17: "and therefore counteracts the development of convection \_in the MBL\_"

22. Pxvii, L20: "... were found at intermediate zenith angles \_for the presented case study\_" However, case study (c) represents a rather thick SAL and so the reported maximum values should hold for long-range transported dust at Barbados.

23. Sometimes your sentences tend to be very long. It would be easier for the reader to split them into two sentences, e.g., Pvii, L21-24 or Pxv, L21-23

References: Burton, S. P., Ferrare, R. A., Hostetler, C. A., Hair, J. W., Rogers, R. R., Obland, M. D., Butler, C. F., Cook, A. L., Harper, D. B., and Froyd, K. D.: Aerosol classification using airborne High Spectral Resolution Lidar measurements – methodology and examples, Atmos. Meas. Tech., 5, 73–98, https://doi.org/10.5194/amt-5-73-2012, 2012.

Groß, S., Freudenthaler, V., Schepanski, K., Toledano, C., Schäfler, A., Ansmann, A., and Weinzierl, B.: Optical properties of long-range transported Saharan dust over Barbados as measured by dual-wavelength depolarization Raman lidar measurements, Atmos. Chem. Phys,15, 11 067–11 080, https://doi.org/10.5194/acp-15-11067-2015,

2015.

Groß, S., Gasteiger, J., Freudenthaler, V., Müller, T., Sauer, D., Toledano, C., and Ansmann, A.: Saharan dust contribution to the Caribbean summertime boundary layer – a lidar study during SALTRACE, Atmos. Chem. Phys, 16, 11 535–11 546, https://doi.org/10.5194/acp-16-35 11535-2016, 2016.

Gutleben, M., Groß, S., Wirth, M., Emde, C., and Mayer, B.: Impacts of water vapor on Saharan Air Layer radiative heating, Geophys. Res. Lett., 46, 14 854–14 862, https://doi.org/10.1029/2019GL085344, 2019.

Haarig, M., Ansmann, A., Gasteiger, J., Kandler, K., Althausen, D., Baars, H., Radenz, M., and Farrell, D. A.: Dry versus wet marine particle optical properties: RH dependence of depolarization ratio, backscatter, and extinction from multiwavelength lidar measurements during SALTRACE, Atmos. Chem. Phys., 17, 14199–14217, https://doi.org/10.5194/acp-17-14199-2017, 2017a.

Haarig, M., Ansmann, A., Althausen, D., Klepel, A., Groß, S., Freudenthaler, V., Toledano, C., Mamouri, R.-E., Farrell, D. A., Prescod, D. A., Marinou, E., Burton, S. P., Gasteiger, J., Engelmann, R., and Baars, H.: Triple-wavelength depolarization-ratio profiling of Saharan dust over Barbados during SALTRACE in 2013 and 2014, Atmos. Chem. Phys., 17, 10767–10794, https://doi.org/10.5194/acp-17-10767-2017, 2017b.

Jung, E., B. Albrecht, J. M. Prospero, H. H. Jonsson, and S. M. Kreidenweis: Vertical structure of aerosols, temperature, and moisture associated with an intense African dust event observed over the eastern Caribbean. J. Geophys. Res. Atmos., 118, 4623–4643, doi:10.1002/jgrd.50352, 2013.

Rittmeister, F., Ansmann, A., Engelmann, R., Skupin, A., Baars, H., Kanitz, T., and Kinne, S.: Profiling of Saharan dust from the Caribbean to western Africa – Part 1: Layering structures and optical properties from shipborne polarization/Raman lidar observations, Atmos. Chem. Phys., 17, 12963–12983, https://doi.org/10.5194/acp-17-

12963-2017, 2017.

Mamouri, R.-E. and Ansmann, A.: Potential of polarization lidar to provide profiles of CCN- and INP-relevant aerosol parameters, Atmos. Chem. Phys., 16, 5905–5931, https://doi.org/10.5194/acp-16-5905-2016, 2016.

---

## Referee Comment (RC2) · Jeffrey Reid (Referee) · 19 Aug 2020

Review of Gutleben et al. This paper presents radiative transfer calculations of Saharan Air Layer (SAL) conditions relative to more background marine demonstrating that dust radiative effects are second order relative to that of water vapor co-transported with the dust. In short, for more background marine conditions large-scale subsidence of dry air is used as a baseline in comparison to SAL dominated environments. Indeed, while the "SAL" is considered dry, it is not as dry as the downward Hadley cell. In concept I think this is a fine exercise and demonstrates the need for more holistic consideration the background aerosol environment. Their finding that differential heating can lead

to SAL destabilization is also interesting, and may explain previous observation that large dust particles are transported further than they theoretically should under laminar conditions (e.g., Maring et al., 2003; something perhaps they may want to mention in their abstract).

In general, I found the paper well written. While there have been simulations done elsewhere to this effect, but I don't think they have been published, certainly not as neatly as this. They also made some changes in response to my "quick" preproposal comments. There are a few things thought at require some correction in a final review. Hope this helps. Jeffrey S. Reid, US Naval Research Laboratory. Introduction

Page 2 line 1: "SALs remain relatively undisturbed and can be transported over thousands of kilometers towards the Caribbean or Americas (Carlson and Prospero, 1972; Karyampudi and Carlson, 1988; Karyampudi et al., 1999)." However, in the PRIDE campaign we showed this is not the case (e.g., Reid et al. 2002,2003; Maring et al 2003) there is considerable variability in dust heights by the time dust reaches the Caribbean. People like the Karyampudi model because it is simple, but it is an idealized situation. Further, the SAL is not always well defined in association with dust transport, especially in January through May. June is a transition month, late July-August is when the Karyampudi SAL model is most appropriate. Indeed, dust transported across the Atlantic in lots of different ways, and the authors language convolutes these mechanisms. During the PRIDE campaign (e.g., Reid et al., 2002), the highest dust concentrations were actually in the marine boundary layer, NOT IN THE SAL. The reasons for this are open to debate (Reid et al., 2003 versus Colarco et al., 2003).

Page 2 Line 13. "We found enhanced water vapor mixing ratios within the SAL compared to the surrounding dry free troposphere." Again, this is true in the context of a well-defined SAL relative to large scale subsidence in the Hadley cell. Under the influence of an easterly wave, there can be quite a lot of moisture around. I have no concern with this study looking at more idealized situations, but it should be mentioned that this analysis is just that, idealized. There is much more variability in the region.

[Figure]

I don't expect the authors to handle this full range of complexity, as their point is well made. But they should discuss it.

Page 3, line 11. "…(NARVAL-II) took place in August 2016…" my point exactly. This is a limited time period in the middle of the most representative of Karyampudi.

Page 5, lien 28. I am not sure why you don't make full use of the HSRL here. We know the mass extinction efficiency for dust is around 0.5 m2 g-1 and you have an extinction measurement. Or if you have noise issues, aerosol backscatter with a mean lidar ratio at least provides linear error propagation (e.g., Reid et al., 2017). Using AERONET retrievals would be a last resort in my mind. In fact, I have my doubts as in mixed environments the retrievals have to apply a mean index of refraction, which does not fit anything.

Page 7, Line 5. As mentioned in my pre review, I think the Hess models have serious problems with dust, right down to incorrect spectral dependence of extinction and large uncertainties in spectral absorption. I think the authors should look hard at the results of Hansel et al. 2009 and Sokolik and Toon 1998. Again, I don't expect the authors to resolve this, and using OPAC is ok for a baseline study. But the authors should be clear on this point.

Page 8, line 8 "hygroscopy" should be hygroscopicity

Page 9, Figure 2. I am not sure based on these lidar profiles that one can say that the MBL goes to 1.6 km. It really depends on where the cloud tops are on whether or not there is detrianmetn there or if it is a residual layer form somewhere else. Mixed layer is easier to define, but MBL top is a bit amorphous.

Page 10/Page 123 line 10/Page 15 line 30: The authors use potential temperature to define mixing, whereas it really should be equivalent potential temperature. Water vapor profiles for case (b) are well mixed in the middle of the SAL, (c) is distinctly not, with multiple water vapor layers visible corresponding with dust concentration. Mixing

ratio should be constant in the presence of mixing. So with the difference in vertical heat shown, why is there stratification? You may want to look at wind shear.

---

## Author Comment (AC1) · 4 Sep 2020

**Author's Response to the Referee Comments**

Manuel Gutleben, Silke Groß, Martin Wirth and Bernhard Mayer

September 1, 2020

The authors would like to thank the referees very much for carefully reading the submitted manuscript and for their helpful and very valuable suggestions and feedbacks. In the following, all comments and questions will be addressed and answered. The comments are repeated and a direct response is given below. In addition, changes in the manuscript are highlighted in the appended marked-up manuscript version using blue (additions) and red (removals) colors.

**Reply to Minor Comments of Referee #2 on 11 June 2020**

**Main points**

**(I) A clear definition of the term '*marine boundary layer*' (MBL) is missing. In the presented case studies (Fig. 4) the so-called MBL reaches up to the trade wind inversion at about 1.4-1.8 km height. Later, it is stated that the SAL is confined at the bottom by the trade wind inversion which is not in line with the indicated SAL in Fig. 4. Please explain in a more quantitative way how the upper limit of the MBL and the lower limit of the SAL are determined in your study. Various attempts to describe the vertical stratification in the Caribbean are found in literature ('convective marine boundary layer (CMBL)' in Groß et al. (2016), 'sub-cloud layer (SCL)' and 'intermediate layer (IL)' in Jung et al. (2013), 'marine aerosol layer (MAL)' in Rittmeister et al. (2017)). The convective part (CMBL or SCL) of the MBL or MAL is clearly visible in Fig. 2 and reaches up to around 600 m height. A definition of the MBL as used in the present study would improve the manuscript.**

Thank you for this valuable feedback. In literature one can find many definitions to partitions of vertical atmospheric profiles in regions impacted by SALs (Figure 1).

Rittmeister et al. (2017) for example, define the so-called MAL (Marine Aerosol Layer) as the layer which reaches from ground-level to the SAL-bottom. In their study they also state that the SAL bottom is equal to the trade wind inversion (TWI). This may be the case for their shipborne lidar observations, but it is certainly not the case for our observations. During NARVAL-II the TWI and the SAL-base differed, as the SAL was often observed to be a decoupled layer which penetrated into the free troposphere (with its own inversions at the top and the bottom). Our measurements indicate three inversions in dust-laden regions (the TWI and two additional ones at the SAL base and the SAL top).

Jung et al. (2013) divide the region below the SAL into a sub-cloud layer (reaching to approximately 500-600 m altitude) and an intermediate layer (IL) that reaches from the SCL top to the SAL bottom. Like Rittmeister et al. (2017) they also identify the SAL base to represent the TWI.

Another approach to characterize the vertical domain below the SAL is performed by Groß et al. (2016). In cloud-free regions they define a convective boundary layer (CMBL) which is confined by strong gradients in particle backscatter, depolarization and potential temperature. In case of a cloudy lower troposphere, the CMBL equals the so-called sub-cloud layer (SCL) and is topped by a cloud layer (CL). The CL itself is confined by an inversion layer.

The dropsonde and lidar analysis presented in Gutleben et al. (2019) partially follows the definitions made by Groß et al. (2016). In their paper the SAL-impacted atmosphere is divided into three regimes:

1. a Marine Boundary Layer (MBL): it is confined by the TWI which is characterized by a rapid temperature decrease and a strong hydrolapse,

2. the Saharan Air Layer (SAL), which is confined by two inversions at the bottom and the top. While the lower inversion is caused by the strong vertical gradients of temperature between the hot SAL-base

[Figure]

Figure 1: Illustration of definitions to partitions of vertical atmospheric profiles in regions impacted by SALs as found in literature.

and the subjacent cooler marine air below (Prospero and Carlson, 1972; Dunion and Velden, 2004), the upper inversion forms due to the predominant large-scale subtropical subsidence in the upper troposphere (Gamo, 1996).

3. a transition or mixed layer between the MBL an the SAL.

As described in Groß et al. (2016), the MBL described in Gutleben et al. (2019) is additionally divided into a sub-cloud layer which extends from the ocean surface to approximately 0.5 to 0.7 km altitude and a cloud layer which extends from the sub-cloud layer top to the TWI. Whereas the sub-cloud layer is well mixed (constant water vapor mixing ratio and constant potential temperature), the cloud layer shows a conditionally unstable lapse rate.

In this paper we follow the definitions made in Gutleben et al. (2019) for the definition of the MBL. They seem to be most appropriate for the observations during NARVAL-II. However, we are aware that a clear description in the submitted manuscript was missing. This is why we added the following paragraph in the revised manuscript:

*Additionally the extent of the lowest atmospheric layer - the marine boundary layer (MBL) - can be determined. It covers the lowest couple of kilometers in a marine atmosphere. The MBL represents a well mixed layer that is characterized by high humidity. As the MBL is confined by the TWI, its upper limit is coming along with a strong increase of potential temperature and a pronounced hydrolapse (Gutleben et al., 2019). By searching for those features in measured profiles of $r_m$ (DIAL) and $\Theta$ (dropsondes), an approximation of the vertical extent of the MBL is additionally outlined.*

**(II) Some comments concerning the aerosol classification (Sec. 2.4.1): You discuss only the contribution of pure mineral dust and pure marine aerosol and its mixture. What about contributions of other aerosol types like pollution from the African continent? At this time of the year, it is not very likely, but should be mentioned. The effect of dry marine aerosol on the depolarization ratio was even studied at Barbados and is described in Haarig et al. (2017a).**

Thank you for this comment. As you already mentioned, the NARVAL-II campaign took place in summer 2016. This time of the year marks the off-season of transatlantic transport of biomass-burning aerosols. No evidence of the transport of biomass burning aerosols towards the Caribbean was found by looking at MODIS imagery along the SAL backward trajectory. Moreover, the particle linear depolarization ratio inside the SAL would decrease significantly with biomass burning aerosols present (e.g. Groß et al., 2013). To clarify this issue we mentioned that no biomass burning aerosols were evident during NARVAL-II in the revised manuscript:
*Other aerosol types, like e.g. African biomass burning aerosols, are also not expected in the measurement region and no evidence for such types was found across the subtropical North Atlantic Ocean by looking at satellite imagery.*
Moreover, we added the study by Haarig et al. (2017a) on dry marine aerosol depolarization to the discussion why no dry marine aerosol is expected at low atmospheric levels. We cited the paper and appended it to the list of references.

**(III) Some comments concerning Section 2.4.2: The lidar ratios for dust (47 sr) and marine aerosol (18 sr) are based on literature from different locations around the globe. However, Groß et al., 2015, reported higher dust lidar ratios of 56 ± 7 sr for Barbados. Could you derive with your high spectral resolution lidar (HSRL) actual lidar ratios of the pure types during your campaign to judge which lidar ratios are more appropriate for your situation? What values for the particle linear depolarization ratio have you used in Eq. 4 and 5? 0.26 and 0.04? From the text it is not clear whether you have calculated the conversion factors (extinction to volume) using the method described in Mamouri and Ansmann (2016) or whether you have taken the values from the literature. Looking into the cited paper, I can't find your conversion factors.**

We followed your suggestion and had a look on the measured lidar ratios during NARVAL-II. We found that the lidar ratios inside dust layers are indeed taking values around 47 sr, and are by far not as high as the presented values by Groß et al. (2015). The differences may arise from the two different measurement methods (Raman in Groß et al. (2015) compared to HSRL in this study). Nevertheless, even if higher lidar ratios for the calculation of the dust-contribution to the total extinction coefficient were used, they would not have a significant impact on the presented results.
The used particle linear depolarization ratios for the conversions are 0.04 for marine aerosol and 0.26 for mineral dust aerosol. We added the information to the revised manuscript.
Thank you for your very useful comment on the used conversion factors. We erroneously mentioned a marine aerosol conversion factor that has not been used for calculations in the submitted manuscript. We used it in a very first approach one or two years ago, before we found out that it is wrong. Both used correct conversion factors for marine and mineral dust aerosol in the submitted study are taken from the study by Groß et al. (2016). In their study they compared in-situ measurements of particle mass concentrations over Barbados to concentrations derived by lidar measurements using conversion factors. They found a good agreement. The used conversion factors are given on page 11538 - chapter 2.7: $0.66 \times 10^{-6}$ m for marine aerosol and $0.65 \times 10^{-6}$ m for transported mineral dust. We are very sorry for the confusion and corrected the factor in the revised manuscript.

**(IV) For your radiative transfer calculations, it is important that no cirrus cloud is present above the scene. The upper panel of Fig. 1 (not yet used in the text) supports the absence of cirrus clouds.**

You are absolutely right. Cirrus clouds would have a major impact on calculated radiative effects and heating rates. We double-checked that issue by looking at MODIS satellite imagery and photos made on-board the HALO aircraft during that flight. As mentioned at the very beginning of Section 3 (Px L09) we took cloudless 5-min-lidar cross sections for calculations. To clarify that the wording '*cloudless*' also implies that no cirrus clouds have been present above the lidar scenes, we added the following sentence to the revised manuscript:
*Here, MODIS imagery of the respective regions is additionally used to ensure that that no cirrus clouds have been present above the observed lidar scenes.*

**Minor points**

**1. Comment: Pii, L12 + L26: Using the term 'we' for studies by the same authors is a matter of taste.**

We changed *we* to *they*.

**2. Comment: Pii, L21: Gutleben et al. (2019x), x = a or b**

We corrected that.

**3. Comment: Pii, L 6: Haarig et al. (2018), describes smoke, you probably mean Haarig et al. (2017b)**

You are right. We corrected this incorrect citation.

**4. Comment: Piv, Eq (1) What is $\delta T/\delta t$? And two lines later a $\delta$ is missing ($\delta F$)**

We corrected the mistakes.

**5. Comment: Pv, L8: Saharan dust *and* marine aerosols**

We added *and*.

**6. Comment: Pv, L11: Burton et al. (2012), is not a good reference for the unchanged dust properties. It describes the aerosol classification and gives just a case study showing mineral dust. Look for other references.**

We removed this citation from the manuscript.

**7. Comment: Pvi, Eq (2)(5) and the whole page: Indices should not be written in italic.**

We changed that.

**8. Comment: Pvii, Table 2: The percentages for the mixing ratio refer to which quantity (volume, number,...)?**

Shown percentages refer to mass mixing ratios. We added that information.

**9. Comment: Pix, Fig. 2, caption: The profiles are not shown in red and blue, but green and black.**

We corrected the text in the figure caption.

**10. Comment: Fig. 2 and Pxi, case (a): There is still a thin layer of some depolarizing particles between 1.7 and 2.5 km height. You are right, there are no dust signatures above 3 km height. But what about the particles above the MBL top? Please discuss this issue shortly.**

We added two sentences to discuss this issue: *Above the MBL some signatures of depolarizing particles ($\delta_{p(532)} < 0.25$) with weak backscatter can be identified. Those signatures are most likely caused by settling dust particles from the dissipating SAL nearby (see Figure 1).*

**11. Comment: Fig. 3, caption: light-green → light-blue**

We corrected the description of the color in the caption.

**12. Comment: Pxi, L27/28: 'Case (a) however, indicates that no distinct correlation of enhanced $r_m$ and $R_{532}$ could be observed in a SAL-free troposphere.' What do you mean by 'no distinct correlation'? Both curves are decreasing with height.**

We removed this confusing sentence and changed it to: *Measured profiles of $r_m$ and $R_{(532)}$ for case (a), show that no enhanced water vapor concentrations could be observed in the SAL-free troposphere. The water vapor profile shows a drop of $r_m$ to values smaller $1\,g\,kg^{-1}$ at altitudes greater $3\,km$, indicating the transition from the MBL to the dry free troposphere.*

**13. Comment: Pxii, L5: 24 h-averaged The explanation follows some pages later: 'It is assumed that the observed profiles do not change and remain stationary within a 24 h time frame.' (Pxiv)**
Here, '*24 h-averaged*' clarifies that heating rates are averaged over a whole day and are given in units $K\,d^{-1}$. However, the description that the observed profiles remain stationary for a 24 h time frame refers to the measurement situation for the calculation of dust radiative effects. To leave no room for misinterpretation we removed this somewhat unnecessary sentence in the revised manuscript:

**14. Comment: Pxii, L11: 2-5 g/kg, one page before you stated 3-5 g/kg**
We corrected that.

**15. Comment: Fig. 4: DIAL measurements are shown in black not blue**
We made the correction.

**16. Comment: Fig. 4 right: Black and dark blue are difficult to distinguish. Please take a different color.**
You are right. Navy blue and black are difficult to distinguish. We changed the colors to blue and grey in the revised manuscript. In our opinion the profiles in the revised figure are way easier to distinguish.

**17. Comment: Sect. 3.3: According to the text, the radiative effects are calculated for the Saharan dust particles disregarding the water vapor. Is this the case? Please state it clearly**
No this is not the case. Calculations of radiative effects are performed using all information from the conducted lidar measurements including measured water vapor concentrations. We clarified that in the revised manuscript.

**18. Comment: Pxv, L8-16: This paragraph should be carefully rewritten. 'Calculations of $RE_{TOA}$ show similar results for case (c). (L10)' - Similar to what? In case of negative numbers, what is considered a minimum? (L13-15)**
Thank you for this very valuable comment. We rewrote the whole paragraph and replaced *maxima and minima* as well as *increasing and decreasing* with *strongest and weakest magnitudes* and *strengthening and weakening*. In this way, the paragraph can be much better understood. The changes in the manuscript can be found in the appended marked-up manuscript version.

**19. Comment: Pxvi, L20: Reference in brackets.**
We removed the brackets.

**20. Comment: Pxvi, L34: January/February 2020**
You are certainly right. The campaign reached into February. We corrected that.

**21. Comment:Pxvii, L17: 'and therefore counteracts the development of convection *in the MBL*'**
We added that.

**22. Comment: Pxvii, L20: '...were found at intermediate zenith angles *for the presented case study*' However, case study (c) represents a rather thick SAL and so the reported maximum values should hold for long-range transported dust at Barbados.**
We followed your suggestion and changed the sentence.

**23. Comment: Sometimes your sentences tend to be very long. It would be easier for the reader to split them into two sentences, e.g., Pvii, L21-24 or Pxv, L21-23**
Following your feedback, we split up several sentences in the revised manuscript. The changes can be found in the marked-up manuscript version.

**Reply to Minor Comments of Referee Jeffrey Reid on 19 August 2020**

**1. Comment: Page 2 line 1: 'SALs remain relatively undisturbed and can be transported over thousands of kilometers towards the Caribbean or Americas (Carlson and Prospero, 1972; Karyampudi and Carlson, 1988; Karyampudi et al., 1999).' However, in the PRIDE campaign we showed this is not the case (e.g., Reid et al., 2002, 2003; Maring, 2003) there is considerable variability in dust heights by the time dust reaches the Caribbean. People like the Karyampudi model because it is simple, but it is an idealized situation. Further, the SAL is not always well defined in association with dust transport, especially in January through May. June is a transition month, late July-August is when the Karyampudi SAL model is most appropriate. Indeed, dust transported across the Atlantic in lots of different ways, and the authors language convolutes these mechanisms. During the PRIDE campaign (e.g., Reid et al., 2002), the highest dust concentrations were actually in the marine boundary layer, NOT IN THE SAL. The reasons for this are open to debate (Reid et al. (2003) versus Colarco et al. (2003)).**

Thank you for this comment. In the framework of your 'Quick Review' you have already pointed out that the Karyampudi-model is an idealized model which mainly applies to the summer months of transatlantic dust transport. We already made changes to the submitted manuscript as a consequence of the quick reviews and highlighted the variability of transatlantic dust transport in the Discussion. However, in our opinion this issue should already be clarified in the Introduction of the paper. This is why we modified the respective paragraph in the Introduction of the revised manuscript and added the information:

*Embedded in the trade wind flow SALs can be transported over thousands of kilometers towards the Caribbean or Americas (Carlson and Prospero, 1972; Karyampudi and Carlson, 1988; Karyampudi et al., 1999). During the summer months from June to August SALs are observed to remain relatively undisturbed during their transatlantic transport. However, previous field campaigns (e.g. the Puerto Rico Dust Experiment PRIDE in 2000), have shown that especially during the winter months dust transport is also occurring at lower atmospheric levels down to sea surface (Reid et al., 2002). Such a transport would modify the aerosol composition inside the boundary layer and would potentially impact the boundary layer state.*

**2. Comment: Page 2 Line 13. 'We found enhanced water vapor mixing ratios within the SAL compared to the surrounding dry free troposphere.' Again, this is true in the context of a well-defined SAL relative to large scale subsidence in the Hadley cell. Under the influence of an easterly wave, there can be quite a lot of moisture around. I have no concern with this study looking at more idealized situations, but it should be mentioned that this analysis is just that, idealized. There is much more variability in the region. I don't expect the authors to handle this full range of complexity, as their point is well made. But they should discuss it.**

You are certainly right. To clarify that, we added some sentences in the Discussion and highlighted that water vapor transport is much more complex and may depend on the synoptic situation and the interaction of the SAL with African Easterly waves:

*Furthermore, the observed vertical water vapor distribution during NARVAL-II may only be representative for an undisturbed SAL-transport during the summer months. Moisture originating from the outflow of the Intertropical convergence zone or from convective systems embedded in African Easterly Waves can modify the vertical moisture distribution in SAL-regions during disturbed transatlantic transports. Questions on the reasons for the variability in the vertical distributions of mineral dust and water vapor as well as whether there is a certain seasonal pattern in these vertical distributions not only at the beginning but also throughout the transatlantic transport can hopefully be answered in near future by analyzing data collected during the recent EUREC$^4$A field campaign (ElUcidating the RolE of Clouds-Circulation Coupling in ClimAte; Bony et al., 2017) in January/February 2020.*

**3. Comment: Page 3, line 11. '. . . (NARVAL-II) took place in August 2016. . .' my point exactly. This is a limited time period in the middle of the most representative of Karyampudi.**

Yes you are right. NARVAL-II provided a snapshot of the SAL-situation during the summer months. An analysis of data collected during the SAL-influenced EUREC$^4$A campaign will provide valuable insights in the Caribbean dust-situation during subtropical winter. It will be interesting to see how the radiative impacts of transported mineral dust during summer (NARVAL-II) and winter ( EUREC$^4$A) differ. We added the following

sentence to the Discussion of the revised manuscript: *An analysis of the EUREC⁴A data set will additionally provide valuable insights on the SAL-radiative effects during subtropical winter months.*

**4. Comment: Page 5, line 28. I am not sure why you don't make full use of the HSRL here. We know the mass extinction efficiency for dust is around $0.5\,\mathrm{m^2 g^{-1}}$ and you have an extinction measurement. Or if you have noise issues, aerosol backscatter with a mean lidar ratio at least provides linear error propagation (e.g. Reid et al., 2017). Using AERONET retrievals would be a last resort in my mind. In fact, I have my doubts as in mixed environments the retrievals have to apply a mean index of refraction, which does not fit anything.**

The calculation of particle mass concentrations using HSRL-extinction measurements together with conversion factors retrieved from the AERONET inversion algorithm is a well known method, which makes full use of the HSRL-capabilities. These factors have even been validated by Groß et al. (2016), who compared in-situ measurements of dust mass concentrations to dust mass concentrations derived from lidar extinction measurements over Barbados. The retrieved mass concentrations from lidar were in good agreement with the concentrations from in-situ measurements. In mixed regimes, we separated the extinction coefficients of mineral dust and marine aerosol, using the method described in Tesche et al. (2009) and Groß et al. (2011). In this way no factors from mixed regimes are applied, but from pure regimes only. We think that this method is the most appropriate method to calculate dust mass concentrations from HSRL-extinction measurements.

**5. Comment: Page 7, Line 5. As mentioned in my pre review, I think the Hess models have serious problems with dust, right down to incorrect spectral dependence of extinction and large uncertainties in spectral absorption. I think the authors should look hard at the results of Hansel et al. 2009 and Sokolik and Toon 1998. Again, I don't expect the authors to resolve this, and using OPAC is ok for a baseline study. But the authors should be clear on this point.**

We are aware that by using inversion techniques together with OPAC, uncertainties are introduced to the results. However, as you already said, this study represents a baseline study and uncertainties in spectral dependencies of extinction have almost no impact on the presented results.
As already mentioned in the response to your 'Quick Reviews' the well-known OPAC by Hess et al. (1998) has been improved within the framework of the recent update by Koepke et al. (2015) (four years after the publication of Hansell et al., 2011). Within this update the transition from calculations based on Mie theory for spherical mineral dust particles to T-Matrix calculations with assumptions on the aspect ratio distribution of prolate spheroids has been made. This change mainly led to improvements for scattering simulations in the solar range, but could also have implications in the comparison of mass extinction efficiencies discussed in Hansell et al. (2011).

**6. Comment: Page 8, line 8 'hygroscopy' should be hygroscopicity.**

We corrected that.

**7. Comment: Page 9, Figure 2. I am not sure based on these lidar profiles that one can say that the MBL goes to 1.6 km. It really depends on where the cloud tops are on whether or not there is detainment there or if it is a residual layer form somewhere else. Mixed layer is easier to define, but MBL top is a bit amorphous.**

Thank you for this comment. Referee #1 already demanded a clear definition of the marine boundary layer (MBL) in this paper. There are many different definitions of the lowermost part of the atmosphere in literature (e.g. Jung et al., 2013; Rittmeister et al., 2017; Groß et al., 2016; Gutleben et al., 2019). As already mentioned in the response to Comment #1 of Referee #1 we follow the MBL-definition made in Gutleben et al. (2019) which seems to be most appropriate for the observations during NARVAL-II: an approximation of the MBL-top is performed by searching for the TWI in dropsonde measurements. It is coming along with a strong hydrolapse and a strong increase of potential temperature.
However, we are aware that in the submitted manuscript a clear statement, that the given MBL and SAL-extents are just approximations from an analysis of dropsonde profiles, is missing so far. We added this information in the text and the figure captions in the revised manuscript. Changes can be found in the marked-up manuscript version.

**8. Comment: Page 10/Page 12 line 10/Page 15 line 30: The authors use potential temperature to define mixing, whereas it really should be equivalent potential temperature. Water vapor profiles for case (b) are well mixed in the middle of the SAL, (c) is distinctly not, with multiple water vapor layers visible corresponding with dust concentration. Mixing ratio should be constant in the presence of mixing. So with the difference in vertical heat shown, why is there stratification? You may want to look at wind shear.**

Thank you for this very valuable comment. We added profiles of equivalent potential temperature ($\Theta_e$) to Figure 4 in the revised manuscript. From surface level to the top of the SALs the profiles of $\Theta_e$ mainly follow the shapes of the profiles of $r_m$. Due to the lack of condensable water vapor in altitudes higher than the SAL-tops, profiles of $\Theta_e$ and $\Theta$ converge with altitude.

We also performed a dropsonde-analysis for profiles of wind speed and wind shear during RF3 to examine the causes for the observed layering of water vapor inside and below the SAL (not shown in the revised manuscript). Therefore, we interpolated the measurements of the 23 deployed dropsondes in this measurement region along the flight track (Figure 2). It can be seen that the vertical layering of water vapor and particle backscattering inside the SAL is coinciding with vertical changes of wind speed (sharp gradients of wind speed at $\sim 3.0\,\mathrm{km}$ and $\sim 4.5\,\mathrm{km}$ altitude). As a result, wind shear is highest at the top of the SAL and at the boundaries of the respective water vapor filaments. In this case, differences in wind shear could be an explanation why vertical mixing processes due to radiative heating are confined to the respective water vapor layers. We added this information to the respective paragraphs in the revised manuscript. Changes can be found in the marked-up manuscript version.

[Figure]

Figure 2: Vertical profiles of wind speed u (right) and wind shear S (left) as measured from dropsondes during RF3 interpolated along the HALO flight path.

[revised manuscript text omitted]